# Immediate Effects of TECAR Therapy on Gastrocnemius and Quadriceps Muscles with Spastic Hypertonia in Chronic Stroke Survivors: A Randomized Controlled Trial

**DOI:** 10.3390/biomedicines11112973

**Published:** 2023-11-04

**Authors:** Laura García-Rueda, Rosa Cabanas-Valdés, Carina Salgueiro, Jacobo Rodríguez-Sanz, Albert Pérez-Bellmunt, Carlos López-de-Celis

**Affiliations:** 1Faculty of Medicine and Health Sciences, Universitat Internacional de Catalunya, 08195 Barcelona, Spain; lgarciar@uic.es; 2Department of Physiotherapy, Faculty of Medicine and Health Sciences, Universitat International de Catalunya, 08195 Barcelona, Spain; carlesldc@uic.es; 3Clínica de Neurorehabilitación Sant Cugat del Vallés, 08195 Barcelona, Spain; carinafsalgueiro@gmail.com; 4Department of Basic Sciences, Faculty of Medicine and Health Sciences, Universitat International de Catalunya, 08195 Barcelona, Spain; aperez@uic.cat; 5ACTIUM Functional Anatomy Group, 08195 Barcelona, Spain; 6Fundació Institut Universitari per a la Recerca a l’Atenció Primària de Salut Jordi Gol i Gurina (IDIAPJGol), 08007 Barcelona, Spain

**Keywords:** Tecar therapy, stroke, spasticity, functional massage, functionality, capacitive–resistive electric transfer therapy, CRet, muscle tone

## Abstract

Background: The aim of this study was to assess changes in muscle properties after a single session of capacitive and resistive energetic transfer (TECAR) therapy on spastic gastrocnemius and quadriceps muscles in chronic post-stroke. Methods: A total of 36 chronic stroke survivors with lower limb hypertonia were enrolled in a double-blind randomized controlled trial. The experimental group (*n* = 18) received a single 30 min session of TECAR therapy in combination with functional massage (FM) on the gastrocnemius and quadriceps muscles. The control group (*n* = 18) received a sham treatment of TECAR therapy (without electrical stimulation) in combination with real FM. The primary outcome was muscle tone of the lower limb muscles assessed with the Modified Ashworth Scale (MAS). The secondary outcomes were goniometric degrees of the MAS (goniometer), neuromuscular properties of the gastrocnemius/quadriceps (myotonometer), and passive range of motion (inclinometer). All measurements were performed at baseline (T0), immediately after treatment (T1), and at 30 min post-treatment (T2) by a blinded assessor. Results: The MAS score ankle dorsiflexion significantly decreased at T0–T1 (*p* = 0.046), and the change was maintained at T0–T2 (*p* = 0.019) in the experimental group. Significant improvements were noted in the passive range of motion for knee flexion (*p* = 0.012) and ankle dorsiflexion (*p* = 0.034) at T2. In addition, knee flexion improved at T1 (*p* = 0.019). Conclusion: A single session of Tecar therapy at the same time with FM on the gastrocnemius and rectus femoris immediately reduces muscle tone and increases the passive range of motion of both ankle and knee in chronic stroke survivors. There were no significant changes in the neuromuscular properties measured with myotonometer.

## 1. Introduction

Spasticity is the most common sequel of stroke. It has a severe impact on motor and functional recovery [1]. There is increased muscle tone due to a lesion of the upper motor neuron that presents as an involuntary, sustained, and intermittent muscle activation [2]. Other related motor impairments derived from stroke coexist with spasticity and share similar pathophysiological origins. These include abnormal synergies and inappropriate and anomalous muscle co-activation [3]. From a biomechanical point of view, the initial post-stroke paresis immobilizes the affected muscles. This leads to adaptive shortening of muscle fibers, the development of soft tissue contractures, and velocity-dependent stretch reflexes [4]. Previous studies suggest that spasticity causes morphological changes in muscle thickness in stroke patients [5]. Changes in soft tissues result in non-neural symptoms, including changes in muscle and tendon properties and reduced joint range of motion (ROM) [6,7].

According to Kuo. et al. [8], the biomechanical component of spasticity may increase over time, although the neurological component (of spasticity) peaks in the third month after stroke. Spasticity is present in 43.2% of stroke survivors at 12 months [9]. The prevalence of increased muscle tone is up to 97% [10] in chronic stroke survivors with moderate to severe motor impairments. It suggests that spontaneous spasticity reduction occurs rarely. It tends to evolve and become more severe over time [11].

Common lower limb muscle involvement of spasticity includes ankle plantar flexors and knee extensors, amongst other muscle groups [10]. Spasticity in gastrocnemius and soleus muscles is very common and often results in various ankle and foot deformities, including equinovarus and toe deformities [12] that impact motor function, gait, and quality of life [13]. Physical therapy interventions, such as massage [14], joint mobilization, and active stretching, are widely used as conservative methods [15] to reduce excess muscle tone and improve motor function. Physical therapy and pharmacological treatment are essential to prevent retraction and joint fixation [16] and improve motor recovery [14]. Functional massage (FM) is a technique that simultaneously combines passive rhythmic mobilization and massage-stretching of the muscles, which reduces excess muscle tone without causing pain [17].

Tecar therapy (TT) is another technique that reduces muscle stiffness [18] and improves flexibility [19]. TT is a modality of non-invasive diathermy that improves superficial and deep blood circulation, hemoglobin saturation, and muscle flexibility by providing high frequency to the tissues [20]. TT is useful in chronic musculoskeletal diseases where an increase in the temperature of deep tissues is needed to change the viscoelasticity of the structures [21]. This effect could be positive in the treatment of hypertonia due to spasticity because its onset and development can be affected by structural changes in muscle and tendon fibers and extracellular components. 

It is important to highlight that secondary musculoskeletal problems can hopefully be prevented, and more effective rehabilitation treatment will optimize recovery [22]. There is a lack of high-quality evidence for many modalities of non-pharmacological interventions for spasticity [23,24]. To the best of our knowledge, there are no randomized controlled trials focused on the effects of TT on hypertonia due to spasticity in stroke survivors [23]. 

Therefore, the aim of this study is to assess changes in muscle tone, passive range of motion, and mechanical properties that occur with a single TT session, in spastic gastrocnemius and quadriceps muscles of chronic stroke survivors, as an adjunct to FM.

## 2. Materials and Methods

### 2.1. Study Design

A randomized double-blind controlled clinical trial was carried out in the laboratory of the Universitat Internacional de Catalunya. This study was approved by the local ethics committee of the university—Research Ethics Committee (FIS-2021-06)—according to the Declaration of Helsinki (World Medical Association, Ferney-Voltaire, France, 2013), and registered at www.clinicaltrials.gov (accessed on 23 March 2023) under number NCT04824768. CONSORT 2010 and TIDieR guidelines were followed.

### 2.2. Sample Size Calculation

The sample size was calculated based on the article by Lee et al. [25] for MAS outcomes. Accepting an alpha risk of 0.05 and a beta risk of 0.2 in a bilateral contrast, a sample of 36 participants (18 in each group) is required to detect a difference of ≥1.13 points in MAS, with a common standard deviation of 1.09 points.

### 2.3. Participants

Chronic stroke volunteers were recruited from different neurorehabilitation clinics and associations in the Catalunya (Spain) area from May 2021 to May 2022.

Inclusion criteria were the following: (1) chronic stroke survivors (6 months post-stroke); (2) age >18 years; (3) score of 1–3 points on the Modified Ashworth Scale (MAS) [26,27] on hip flexion, knee flexion, or ankle dorsiflexion; and (4) score >25 points on the Montreal Cognitive Assessment [28]. Exclusion criteria were the following: (1) injuries in the lower limbs; (2) other neurological diseases or cancer; (3) osteosynthetic material or pacemaker; (4) botulinum toxin or antispastic treatment 3 months prior to the study; (5) inability to remain in the prone position; and (6) any contraindication to massage and Tecar not mentioned in the exclusion criteria such as skin infections, inflammatory vascular diseases, or acute inflammation. The patients signed an informed consent form after receiving written and verbal descriptions of the study procedures. 

### 2.4. Randomization

The randomization was performed with OxMar computer software by a researcher who was not involved in data collection. The allocation concealment of the sequence was performed with opaque and numbered envelopes. Participants were randomly assigned to two groups. Both groups received FM with the mobile resistive/capacitive electrodes, and at the same time, the experimental group received TT, and the control group received the sham TT. The TT treatment, in combination with FM, was applied to the quadriceps, gastrocnemius, and Achilles tendon of the affected leg with the T-plus device (Wintecare Chiasso, Switzerland).

### 2.5. Intervention

The experimental group received the procedure and dosage detailed below. The participant was in a prone position. A fixed electrode was set under the participant’s abdomen. The protocol started with massage and TT in the resistive mode (80–100 W, 7 min) in the lumbar area, followed by massage and TT in the resistive mode (100–120 W, 5 min) in the hamstrings of the affected leg. We continued in the gastrocnemius area with FM and TT in the same resistive mode (110–120 W, 5 min), followed by FM and TT in the capacitive mode in the same region (180–200 VA, 4 min).

The patient was moved to a supine position with the electrode fixed on the lumbar area. FM and TT in the resistive mode (110–140 W, 5 min) were performed on the quadriceps, ending the session with FM and TT in the capacitive mode in the same region (180–200 VA, 4 min). The session lasted approximately 30 min. A “Thermocomed” digital thermometer (precision ±0.3° Celsius) was used to measure the surface temperature of the affected lower limb during the treatment.

The same steps of the experimental group were followed for the control group but with sham TT (W/VA). To blind the participant to whether the electrode was generating a dose, we preheated it to provide a slight cutaneous thermal sensation (Beurer HK Comfort heat mat). A blinded assessor (RN) collected and recorded the measurements. All outcomes were measured at baseline (T0), immediately after the intervention (T1), and 30 min at the end-intervention (T2). The entire procedure lasted approximately 90 min. Although TT can significantly increase skin temperature [21], the doses applied were previously established by a physiotherapist experienced with TT. Room temperature was controlled at 22–23° to prevent an increase in muscle tone.

### 2.6. Outcomes

Nominal and clinical data were recorded prior to the measurement of motor functions. The outcome assessment procedure was carried out in the following order.

#### 2.6.1. Neuromuscular Properties (Myotonometry)

Muscle properties of gastrocnemius and quadriceps muscles were assessed by myotonometry with MyotonPro (Myoton Ltd., Tallinn, Estonia) [29]. The MyotonPro assesses tone or state of tension natural oscillation frequency (Hz), biomechanical properties as dynamic stiffness (N/m), and viscoelastic properties as mechanical stress relaxation time (ms) [30]. The MyotonPro device was placed on the skin, perpendicular to the surface of the most prominent belly of the medial and lateral gastrocnemius and quadriceps muscles, with the participant in the supine position (Figure 1A,B).

#### 2.6.2. Passive Range of Motion (PROM)

Passive range of motion (PROM) at the end-feel of the ankle and knee was measured using an inclinometer (Clinometer Smartphone Application TM 4.9.2). To determine the strength, a handheld dynamometer (MicroFET2, Hoggan Scientific, Salt Lake City, UT, USA) was used. The participant rested in a supine position with the knee straight on the treatment table to measure the ankle PROM. The assessor applied the maximum force against the metatarsal base with the dynamometer to assess at T0 (Figure 2A), and the assessor applied the inclinometer along the fifth metatarsal bone (Figure 2B). Assessing at T1 and T2, the investigator applied the same force registered by the inclinometer at T0. Measuring knee PROM, the participant rested in a supine position, close to the edge of the treatment table, with the knee bent as the leg was outside the treatment table. For T0, the investigator applied the maximum force against the distal third of the tibial shaft, and the evaluator applied the inclinometer along the proximal shaft (Figure 2C). For T1 and T2, the investigator applied the same force registered by the inclinometer at T0.

#### 2.6.3. Modified Ashworth Scale (MAS)

Muscle tone was assessed with MAS. It is the most universally accepted clinical tool used to measure increases in muscle tone [31]. The MAS is a 6-point scale. Scores range from 0 to 4, where lower scores represent normal muscle tone and higher scores represent spasticity or increased resistance to passive movement. For the gastrocnemius muscle, the hip was in 45 degrees of flexion with the knee in maximum extension, and the ankle was moved from maximum plantar flexion to maximum dorsiflexion. For the rectus femoris muscle, the knee and hip were in maximal extension, and the knee was moved from maximum extension to maximum flexion.

#### 2.6.4. Degrees of Modified Ashworth Scale

Degrees of the MAS for hip flexion, knee flexion, and ankle dorsiflexion were assessed with a long arm universal goniometer (Enraf Nonius, Prim Group, Madrid, Spain). The participant was lying in lateral decubitus position, with hip and knee at 0°, and then the assessor slowly moved their leg to maximum passive hip flexion, knee flexion, and ankle dorsiflexion. The assessor performed the respective measurements with the goniometer.

### 2.7. Data Analysis

Statistical analysis was performed with IBM SPSS Statistic version 26.0 (Armonk, NY, USA: IBM Corp) to assess group differences in the variables at each time interval. A descriptive analysis was conducted. The mean and standard deviation were calculated for the quantitative variables. Frequencies were calculated for demographic and anthropologic qualitative variables. The Shapiro–Wilk test was used to determine non-normal distribution of quantitative data.

A repeated-measures analysis of variance (ANOVA) with time (baseline, post-intervention, and follow-up) and group (experimental and control) was conducted to determine changes in the outcomes of each dependent variable (MAS score, MAS degrees, PROM of knee flexion and ankle dorsiflexion, and neuromuscular properties of gastrocnemius and rectus femoris muscles) at each time interval. If the assumption of the sphericity test was not satisfied, the Greenhouse–Geisser correction was used for interpretation. When a statistically significant effect was observed, a post hoc analysis was performed, and the Bonferroni correction was used to adjust for multiple comparisons. For the qualitative variable, MAS, McNeimar’s test was used for the within-group analysis, and Fisher’s exact statistic for the between-group analysis.

All individuals originally enrolled were included in the final analysis as planned. Effect sizes were calculated using eta squared (ŋ2), considering an effect size of >0.14 as large, around 0.06 as medium, and <0.01 as small. The level of significance was set at *p* < 0.05.

## 3. Results

Of 44 volunteers (16 females and 28 males) recruited, 2 females and 6 males did not meet the inclusion or exclusion criteria. Two participants scored >2, and one scored <1 on the Modified Ashworth Scale. One participant carried osteosynthetic material that was incompatible with Tecar therapy. The remaining three excluded participants were in the subacute phase of stroke. Thus, this study involved 36 participants (18 experimental group and 18 control group). The mean age was 58.6 ± 11.3 years. There were no dropouts for the 90 min after measurements. There were no significant differences between the two groups for any demographic or baseline measures (Table 1). The comparative analyses of this study can be found in Table 2, Table 3 and Table 4.

The MAS score for ankle dorsiflexion showed statistically significant changes compared to the control group between T0 and T1 (*p* = 0.046) and between T0 and T2 (*p* = 0.019). No statistically significant changes were observed for MAS hip flexion or MAS knee flexion (Table 2).

The MAS degrees for ankle dorsiflexion showed significant improvements for the between-group analysis between T2 and T0 (*p* = 0.011) (Table 3). In the within-group analysis, significant improvements were noticed for the experimental group between T2 and T0 (*p* = 0.004) (Table 4). For MAS degrees, knee flexion showed significant effects for between-group analysis between T1 and T0 (*p* = 0.016) and between T2 and T0 (*p* = 0.000) (Table 3). In the within-group analysis, statistically significant differences were observed in the experimental group, between T0 and T1 (*p* = 0.005) and between T0 and T2 (*p* = 0.002). (Table 4). The MAS degrees for hip flexion showed significant main effects in the between-group analysis between T2 and T0 periods (*p* = 0.022) (Table 3). In the within-group analysis, statistically significant differences were noticed in the experimental group, in the T0–T2 (*p* = 0.003) (Table 4).

In the between-group analysis of the PROM of the gastrocnemius variable, statistically significant differences were demonstrated between T2 and T0 (*p* = 0.034). In the within-group analysis, significant improvement was observed in the experimental group between T1 and T0 (*p* = 0.028) and between T2 and T0 (*p* = 0.033) (Table 3 and Table 4). In the PROM of the quadriceps variable, statistically significant differences were found in the between-group analysis between T1 and T0 (*p* = 0.012) and between T2 and T0 (*p* = 0.019). In the within-group analysis, no statistically significant changes were observed in both groups (Table 3 and Table 4).

Regarding the neuromuscular properties of the gastrocnemius and quadriceps assessed with myotonometry, there were no improvements in the tone of gastrocnemius medialis/lateralis or quadriceps, neither in the between-group nor within-group analysis in both groups (Table 3 and Table 4). In the between-group analysis, no statistically significant changes were found for gastrocnemius medialis in any time assessment. In the within-group analysis, statistically significant changes were revealed for gastrocnemius medialis in the experimental group between T1 and T0 (*p* = 0.021). There were no significant improvements in the stiffness of gastrocnemius lateralis or quadriceps, neither in the between-group nor within-group analysis in both groups (Table 3 and Table 4). There were no significant improvements in the relaxation of gastrocnemius medialis/lateralis or quadriceps, neither in the between-group nor within-group analysis in any group (Table 3 and Table 4). No adverse events were observed with the interventions performed in this study.

## 4. Discussion

The results of this study suggest that a single session of TT in combination with FM immediately reduces the muscle tone of the gastrocnemius due to spastic hypertonia in chronic stroke survivors. A reduction in the MAS score and an increase in the MAS degrees for ankle dorsiflexion in the experimental group was observed. Moreover, the PROM at the end-feel of the gastrocnemius also increased in the experimental group.

There were no changes in the MAS score for knee and hip flexion, although the MAS degrees increased for knee and hip flexion in the experimental group, which indicates an improvement in the tone of the rectus femoris muscle. Furthermore, the PROM at the end-feel of the knee flexion increased between T1 and T0 and T2 and T0. However, no changes were observed in neuromuscular properties (tone, stiffness, and relaxation), assessed by myotonometry, except for gastrocnemius medialis stiffness in the experimental group at the end of the intervention.

To the best of our knowledge, the effects of TT, in addition to FM, have not been assessed on the neuromuscular properties of post-stroke spastic muscles in previous studies. Therefore, it has not been possible to compare our results with other similar studies. Although the MAS is the most accepted clinical tool that measures resistance to passive movement due to excess muscle tone, it has shown several limitations since it is a subjective method based on the experience of the assessor [31,32]. However, it is the most universally accepted tool to measure the increase in muscle tone in clinical practice and research.

We have included the MAS degree variable in this study. It has provided additional objective information regarding muscle tone. Both the MAS score and the MAS degrees of ankle dorsiflexion improved in the experimental group; however, although the MAS score of the knee and hip flexion did not change, the MAS degrees showed improvement in the experimental group. Vidmar et al. [33] showed that the intrarater reliability of the MAS is moderate to excellent for lower limb muscles, which could be in line with our results. This would suggest that other complementary assessments could contribute to improving objective information to the MAS regarding muscle tone.

Pre- and post-treatment findings revealed no differences for the control group for any of the variables in this study. Only the experimental group showed significant improvements in MAS score, MAS degrees, and PROM, so the combination of TT with FM could exert more changes in muscle properties than FM alone.

Stretching appears to have no benefit for neurological disorders like stroke or spinal cord injury [34]. It is not surprising that simple muscle stretching techniques are not effective. Different techniques of manual therapy are still considered basic approaches in most rehabilitation programs, and they are usually used in isolation or associated with conventional physiotherapy [14,35,36,37]. Therapeutic massage, in addition to conventional therapy, is considered an effective option to treat post-stroke spastic hypertonia, as it can help to optimize the sarcomere’s length [38] and to increase the range of motion and flexibility [14,39]. A study by Bïngol et al. [37] detected that FM was effective in reducing lower limb spasticity in children with cerebral palsy. This study was conducted over eight weeks, with two 45 min sessions per week, whereas our study consisted of a unique session. Possibly, no significant changes were observed in the control group for any outcomes, likely because only one session was conducted. However, as Szabo et al. [36] suggested, manual therapy combined with TT provides more effective muscle stimulation in all types of muscle diseases. Therefore, TT with FM could also exert changes on the biomechanical component of spastic hypertonia in chronic stroke survivors. Some studies hypothesize that not only hyperexcitability reflexes but also altered properties of muscle tissue cause resistance to passive movement [40]. In our study, the improvement in gastrocnemius hypertonia may be due to the quick action on intramuscular blood flow of both the capacitive and the resistive modes of TT. Additionally, TT improves blood circulation in the peritendinous region [41]. The capacitive mode generates heat in more superficial tissues rich in water, such as the gastrocnemius. The resistive mode affects denser and more fibrous tissues, such as the Achilles tendon. In post-stroke hypertonia, there is a morphological disorganization of the collagen fibers of the Achilles tendon, which leads to tendon thickening and reduced tendon flexibility [42].

Our study revealed fascicle lengthening of the calf muscles associated with the increased ankle joint range of motion. Different results were obtained in the knee where only the MAS degrees, but not the MAS score, for hip and knee flexion improved in the experimental group. These results could be explained by the greater thickness and extension of the quadriceps compared to the calf muscles, the latter obtaining greater benefits with the same treatment time. Furthermore, less muscle mass and more intramuscular fat have been recently found in the paretic quadriceps of chronic stroke survivors [43,44]. TT targets tissues with more fat and adipose matter [45,46], which could have an effect on the muscular properties of the rectus femoris in chronic stroke survivors, but not in our participants: there were no changes related to tone, stiffness, and relaxation measured by myotonometry, although a tendency towards improvement was detected in gastrocnemius medialis stiffness at the end of intervention. Akazawa et al. [47] suggest that decreasing intramuscular fat and increasing muscle mass of the quadriceps may improve muscle strength in chronic stroke survivors. However, as Adringa et al. [48] indicate, spasticity reduction is not always accompanied by functional improvement. Thus, we hypothesize that not only more sessions of TT plus FM may improve lower limb functionality in chronic stroke survivors but also a combination of this treatment with an exercise program.

Spasticity, disuse, and lack of activity following stroke produce adaptive changes in the biomechanics, anatomy, and functionality of the musculoskeletal and nervous systems. According to Bavikatte et al. [49], early detection and management of spasticity would improve function and independence and avoid long-term complications. Therefore, TT plus FM could help to improve function by increasing range of motion and improving muscle tone in the subacute stage of stroke, also associated with an exercise program [50]. 

It is also important to highlight that TT generates a thermotherapeutic effect in deep muscle layers, joints, and tendons without excessively increasing skin temperature. In this respect, TT was highly tolerable for our patients. On the other hand, a prospective study by Wissel et al. [51] reported that spasticity is associated with pain in the upper limb. Therefore, applying TT plus FM on the upper limb could help to reduce spasticity and pain in the paretic arm of chronic stroke survivors.

This study has limitations. Firstly, the sample is too small to draw general conclusions. Secondly, we only provided results of the immediate effects of a single session. The reason was to have more control over factors that could mask the technique. This was achieved by continuous monitoring of the subjects. We also wanted to know whether there were any adverse effects with this first study. However, more studies in usual clinical conditions are needed. Finally, as the effects of TT plus FM were analyzed in chronic stroke survivors, we could not collect information regarding the subacute phase.

Further studies are needed, with a larger sample and a greater number of sessions, to be able to fully assess the effects of TT on the neuromuscular properties of chronic stroke survivors. In future studies, it would be interesting to consider the subacute stage of stroke, as well as to conduct the study in the upper limb, where the prevalence of post-stroke spasticity is higher.

## 5. Conclusions

According to our results, a single session of TT in combination with FM can immediately increase ankle and knee passive range of motion and reduce muscle tone of gastrocnemius and rectus femoris muscles in chronic stroke survivors. No adverse events were observed in the application of TT plus FM, and the participants rated the treatment as satisfactory.

## Figures and Tables

**Figure 1 biomedicines-11-02973-f001:**
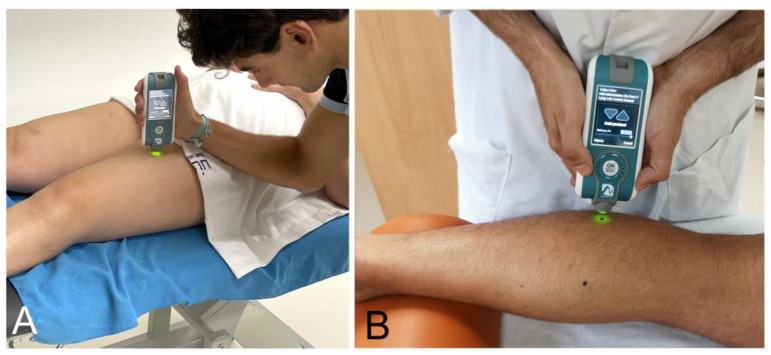
(**A**) Myotonometry on the rectus femoris (quadriceps); (**B**) myotonometry on gastrocnemius lateralis.

**Figure 2 biomedicines-11-02973-f002:**
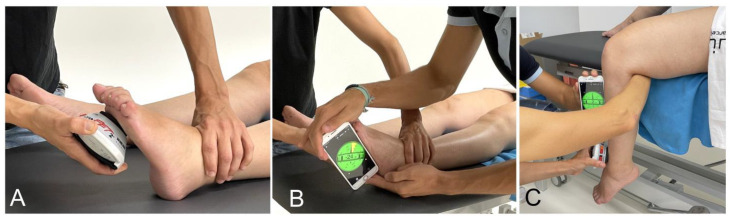
(**A**) Dynamometer against the metatarsal base; (**B**) inclinometer on the 5th metatarsal bone; (**C**) inclinometer on the proximal tibial shaft.

**Table 1 biomedicines-11-02973-t001:** Descriptive characteristics of the participants.

	Experimental Group	Control Group
	Mean ± SD *n* (%)	Mean ± SD *n* (%)
Sex		
Women	7 (38.9%)	7 (38.9%)
Men	11 (61.1%)	11 (61.1%)
Age (years)	58.8 ± 11.9	58.3 ± 11.0
Weight (kg)	75.3 ± 14.6	76.6 ± 17.8
Height (cm)	169.4 ± 9.8	170.6 ± 7.0
BMI	26.1 ± 4.1	26.3 ± 5.6
Type of stroke		
Hemorrhagic	9 (50%)	6 (33.3%)
Ischemic	9 (50%)	12 (55.7%)
Time onset (years)	6.4 ± 3.1	11.1 ± 8.4
Affected side		
Right	14 (77.8%)	10 (55.6%)
Left	4 (22.2%)	8 (44.4%)
Tobacco/alcohol use		
None	14 (77.8%)	11 (61.1%)
Tobacco	1 (5.6%)	4 (22.2%)
Alcohol	3 (16.7%)	3 (16.7%)
Physiotherapy (days/week)	1.6 ± 1.0	2.2 ± 2.5

Abbreviations: BMI, body mass index; SD, standard deviation; *n*, number; kg, kilograms; cm, centimeters.

**Table 2 biomedicines-11-02973-t002:** Descriptive data of MAS score and within-group significance at different times.

		T0	T1	T2	T0–T1	T0–T2	T1–T2
		*n* (%)	*n* (%)	*n* (%)	*p*	*p*	*p*
Experimental Group							
MAS Hip flexion	0 Normal	10 (55.6%)	11 (61.1%)	11 (61.1%)	0.368	0.368	1.000
	1 Light tone	5 (27.8%)	5 (27.8%)	5 (27.8%)			
	1+ Light tone plus	3 (16.7%)	2 (11.1%)	2 (11.1%)			
	2 Pronounced tone	-	-				
MAS Knee flexion	0 Normal	4 (22.2%)	8 (44.4%)	8 (44.4%)	0.136	0.199	0.607
	1 Light tone	8 (44.4%)	6 (33.3%)	5 (27.8%)			
	1+ Light tone plus	2 (11.1%)	3 (16.7%)	3 (16.7%)			
	2 Pronounced tone	4 (22.2%)	1 (5.6%)	2 (11.1%)			
MAS Ankle dorsiflexion	0 Normal	-	-	-	0.046	0.019	0.261
	1 Light tone	1 (5.6%)	5 (27.8%)	5 (27.8%)			
	1+ Light tone plus	4 (22.2%)	6 (33.3%)	7 (38.9%)			
	2 Pronounced tone	13 (72.2%)	7 (38.9%)	6 (33.3%)			
Control Group							
MAS Hip flexion	0 Normal	11 (61.1%)	11 (61.1%)	11 (61.1%)	1.000	1.000	0.317
	1 Light tone	5 (27.8%)	6 (33.3%)	5 (27.8%)			
	1+ Light tone plus	1 (5.6%)	1 (5.6%)	2 (11.1%)			
	2 Pronounced tone	1 (5.6%)	-	-			
MAS Knee flexion	0 Normal	6 (33.3%)	6 (33.3%)	6 (33.3%)	1.000	1.000	1.000
	1 Light tone	7 (38.9%9	7 (38.9%)	7 (38.9%)			
	1+ Light tone plus	3 (16.7%)	3 (16.7%)	3 (16.7%)			
	2 Pronounced tone	2 (11.1%)	2 (11.1%)	2 (11.1%)			
MAS Ankle dorsiflexion	0 Normal	3 (16.7%)	3 (16.7%)	4 (22.2%)	0.368	0.261	0.368
	1 Light tone	4 (22.2%)	5 (27.8%)	5 (27.8%)			
	1+ Light tone plus	3 (16.7%)	3 (16.7%)	2 (11.1%)			
	2 Pronounced tone	8 (44.4%)	7 (38.9%)	7 (38.9%)			

Abbreviations: MAS, Modified Ashworth Scale; T0–T1–T2, time pre–after–30 min after the intervention; *n*, number; *p*, level of significance.

**Table 3 biomedicines-11-02973-t003:** Between-group analysis of the variables MAS degrees, PROM of knee and ankle dorsiflexion, and neuromuscular properties of gastrocnemius and quadriceps.

	Difference T1–T0			Difference T2–T0			Difference T2–T1		
Variable	Experimental Group	Control Group		Experimental Group	Control Group		Experimental Group	Control Group	
	Mean ± SD	Mean ± SD	*p*	Mean ± SD	Mean ± SD	*p*	Mean ± SD	Mean ± SD	*p*
MAS Hip (°)	5.3 ± 9.4	3.2 ± 6.0	0.443	8.0 ± 8.4	1.6 ± 6.5	0.022	2.7 ± 8.6	−1.7 ± 6.9	0.542
MAS Knee (°)	11.3 ± 12.7	3.3 ± 6.2	0.016	11.9 ± 12.4	1.6 ± 4.1	0.000	0.7 ± 5.7	−1.8 ± 5.1	0.239
MAS Ankle (°)	2.2 ± 4.5	1.7 ± 3.2	0.501	3.4 ± 3.7	1.1 ± 2.7	0.011	1.2 ± 3.7	−0.6 ± 3.5	0.134
PROM—Ankle dorsiflexion (°)	2.9 ± 4.2	0.4 ± 3.1	0.161	3.2 ± 4.7	0.3 ± 2.8	0.034	0.3 ± 1.6	−0.1 ± 1.8	0.888
PROM—Knee (°)	2.5 ± 9.6	1.0 ± 3.0	0.012	2.9 ± 9.4	1.1 ± 3.0	0.019	0.4 ± 2.0	0.2 ± 0.9	0.323
GM—Tone (Hz)	4.1 ± 22.9	−0.6 ± 2.7	0.388	−1.5 ± 3.2	−0.4 ± 2.0	0.186	−5.7 ± 23.7	0.3 ± 3.1	0.300
GM—Stiffness (N/m)	−36.8 ± 50.8	−19.7 ± 50.5	0.317	−28.5 ± 73.4	−5.9 ± 52.9	0.297	8.3 ± 61.6	13.8 ± 65.6	0.799
GM—Relaxation (m/s)	2.6 ± 4.3	0.8 ± 4.2	0.212	1.9 ± 5.3	0.4 ± 4.2	0.344	−0.7 ± 5.2	−0.5 ± 4.3	0.863
GL—Tone (Hz)	−1.2 ± 2.8	−0.8 ± 3.6	0.664	−1.6 ± 2.7	−0.1 ± 3.8	0.198	−0.3 ± 2.6	0.6 ± 3.7	0.373
GL—Stiffness (N/m)	−20.3 ± 46.6	−36.1 ± 102.8	0.556	−26.2 ± 44.0	−21.3 ± 132.7	0.884	−5.9 ± 44.6	14.8 ± 80.2	0.346
GL—Relaxation (m/s)	2.2 ± 5.0	1.5 ± 4.7	0.661	2.5 ± 4.4	−0.5 ± 5.6	0.084	0.3 ± 3.7	−2.0 ± 4.7	0.118
RF—Tone (Hz)	−0.2 ± 1.3	0.0 ± 1.8	0.682	−0.2 ± 1.3	0.0 ± 1.2	0.638	0.0 ± 1.0	0.0 ± 1.6	0.957
RF—Stiffness (N/m)	−4.3 ± 23.9	0.8 ± 53.5	0.710	−7.6 ± 22.4	−2.8 ± 28.0	0.568	−3.3 ± 18.0	−3.6 ± 45.8	0.979
RF—Relaxation (m/s)	−2.5 ± 11.2	−16.8 ± 70.3	0.401	−1.8 ± 11.2	−16.8 ± 69.9	0.376	0.7 ± 1.8	0.0 ± 2.4	0.332

Abbreviations: MAS, Modified Ashworth Scale; T0–T1–T2, time pre–after–30 min after the intervention; SD, standard deviation; *p*, level of significance; (°), degrees; GM, gastrocnemius medialis; GL, gastrocnemius lateralis; RF, rectus femoris.

**Table 4 biomedicines-11-02973-t004:** Within-group analysis of the variables MAS degrees, PROM of knee and ankle dorsiflexion, and neuromuscular properties of gastrocnemius and quadriceps.

	T0	T1	Difference T1–T0				T2	Difference T2–T0				Difference T2–T1			
Variables	Mean ± SD	Mean ± SD	Mean	95% CI	*p*	ŋ^2^	Mean ± SD	Mean	95% CI	*p*	ŋ^2^	Mean	95% CI	*p*	ŋ^2^
Experimental Group															
MAS Hip (°)	97.6 ± 21.1	102.9 ± 18.2	5.3	[0.62; 11.18]	0.089	0.02	105.6 ± 18.2	8.0	[2.73; 13.27]	0.003	0.04	2.7	[−2.66; 8.10]	0.590	0.01
MAS Knee (°)	104.7 ± 16.1	115.9 ± 16.2	11.3	[3.31; 19.25]	0.005	0.11	116.6 ± 14.8	11.9	[4.20; 19.69]	0.002	0.13	0.7	[−2.88; 4.21]	1.000	0.00
MAS Ankle (°)	18.8 ± 5.6	20.9 ± 7.3	2.2	[−0.66; 4.99]	0.173	0.03	22.2 ± 6.8	3.4	[1.07; 5.71]	0.004	0.07	1.2	[−1.08; 3.52]	0.528	0.01
PROM—Ankle dorsiflexion (°)	14.0 ± 6.9	16.8 ± 8.7	2.9	[0.27; 5.48]	0.028	0.03	17.1 ± 8.8	3.2	[0.22; 6.09]	0.033	0.04	0.3	[−0.73; 1.28]	1.000	0.00
PROM—Knee (°)	28.1 ± 15.8	30.6 ± 12.6	2.5	[−3.49; 8.39]	0.850	0.01	31.1 ± 12.6	2.9	[−2.99; 8.82]	0.621	0.01	0.4	[−0.84; 1.67]	1.000	0.00
GM—Tone (Hz)	17.8 ± 3.7	21.9 ± 22.6	4.1	[−10.19; 18.45]	1.000	0.02	16.3 ± 3.8	−1.5	[−3.54; 0.45]	0.165	0.04	−5.7	[−20.52; 9.18]	0.975	0.03
GM—Stiffness (N/m)	323.4 ± 71.3	286.6 ± 46.6	−36.8	[−68.62; −5.02]	0.021	0.09	294.9 ± 67.9	−28.5	[−74.42; 17.43]	0.354	0.04	8.3	[−30.19; 46.84]	1.000	0.00
GM—Relaxation (m/s)	18.4 ± 4.6	21.1 ± 6.0	2.6	[−0.05; 5.32]	0.056	0.08	20.3 ± 6.1	1.9	[−1.40; 5.21]	0.434	0.03	−0.7	[−3.97; 2.50]	1.000	0.00
GL—Tone (Hz)	18.3 ± 4.2	17.1 ± 4.6	−1.2	[−3.00; 0.53]	0.240	0.02	16.8 ± 4.6	−1.6	[−3.26; 0.13]	0.075	0.03	−0.3	[−1.98; 1.36]	1.000	0.00
GL—Stiffness (N/m)	352.5 ± 101.0	332.2 ± 95.9	−20.2	[−49.42; 8.88]	0.247	0.01	326.3 ± 94.1	−26.2	[−53.68; 1.36]	0.066	0.02	−5.9	[−33.82; 22.04]	1.000	0.00
GL—Relaxation (m/s)	17.4 ± 5.6	19.6 ± 6.7	2.2	[−0.86; 5.35]	0.217	0.03	19.9 ± 6.6	2.5	[−0.20; 5.24]	0.075	0.04	0.3	[−2.01; 2.57]	1.000	0.00
RF—Tone (Hz)	14.5 ± 1.8	14.3 ± 1.6	−0.2	[−1.05; 0.58]	1.000	0.00	14.3 ± 1.6	−0.2	[−0.99; 0.59]	1.000	0.00	0.0	[−0.61; 0.68]	1.000	0.00
RF—Stiffness (N/m)	287.5 ± 36.8	283.2 ± 35.7	−4.3	[−19.29; 10.61]	1.000	0.00	279.9 ± 38.9	−7.6	[−21.64; 6.34]	0.495	0.01	−3.3	[−14.56; 7.95]	1.000	0.00
RF—Relaxation (m/s)	23.9 ± 12.0	21.4 ± 3.7	−2.5	[−9.52; 4.49]	1.000	0.02	22.1 ± 3.5	−1.8	[−0.46; 1.83]	1.000	0.01	0.7	[−0.46; 1.86]	0.398	0.01
Control Group															
MAS Hip (°)	100.4 ± 15.8	103.6 ± 16.0	3.2	[−0.55; 6.99]	0.110	0.01	101.9 ± 18.6	1.6	[−2.49; 5.60]	0.965	0.00	−1.7	[−5.96; 2.63]	0.951	0.00
MAS Knee (°)	100.9 ± 22.1	104.3 ± 21.8	3.3	[−0.56; 7.23]	0.109	0.01	102.5 ± 22.0	1.6	[−1.02; 4.13]	0.383	0.00	−1.8	[−4.95; 1.40]	0.466	0.00
MAS Ankle (°)	18.8 ± 6.4	20.5 ± 7.8	1.7	[−0.35; 3.68]	0.126	0.01	19.9 ± 7.2	1.1	[−0.65; 2.77]	0.359	0.01	−0.6	[−2.81; 1.59]	1.000	0.00
PROM—Ankle dorsiflexion (°)	16.7 ± 5.7	17.1 ± 6.6	0.4	[−1.58; 2.36]	1.000	0.00	17.0 ± 6.5	0.3	[−1.48; 2.03]	1.000	0.00	−0.1	[−1.22; 1.00]	1.000	0.00
PROM—Knee (°)	25.4 ± 16.7	26.3 ± 17.1	1.0	[−0.93; 2.84]	0.588	0.00	26.5 ± 16.8	1.1	[−0.73; 2.98]	0.380	0.00	0.2	[−0.37; 0.70]	1.000	0.00
GM—Tone (Hz)	17.2 ± 3.6	16.6 ± 3.1	−0.6	[−2.34; 1.09]	1.000	0.01	16.9 ± 3.3	−0.4	[−1.59; 0.88]	1.000	0.00	0.3	[−1.65; 2.19]	1.000	0.00
GM—Stiffness (N/m)	305.4 ± 81.0	285.8 ± 62.5	−19.7	[−51.27; 11.94]	0.350	0.02	299.6 ± 72.4	−5.9	[−38.98; 27.20]	1.000	0.00	13.8	[−27.28; 54.83]	1.000	0.01
GM—Relaxation (m/s)	18.3 ± 5.8	19.2 ± 5.0	0.8	[−1.82; 3.47]	1.000	0.01	18.7 ± 5.8	0.4	[−2.27; 3.02]	1.000	0.00	−0.5	[−3.17; 2.26]	1.000	0.00
GL—Tone (Hz)	16.8 ± 4.2	16.1 ± 3.5	−0.8	[−3.00; 1.47]	1.000	0.01	16.7 ± 3.3	−0.1	[−2.49; 2.25]	1.000	0.00	0.6	[−1.69; 2.98]	1.000	0.01
GL—Stiffness (N/m)	326.7 ± 113.7	290.6 ± 57.7	−36.1	[−100.46; 28.24]	0.464	0.04	305.4 ± 86.4	−21.3	[−104.37; 61.71]	1.000	0.01	14.8	[−35.40; 64.95]	1.000	0.01
GL—Relaxation (m/s)	19.0 ± 6.0	20.6 ± 5.2	1.5	[−1.42; 4.48]	0.560	0.02	18.6 ± 4.3	−0.5	[−3.95; 3.04]	1.000	0.00	−2.0	[−4.94; 0.97]	0.973	0.04
RF—Tone (Hz)	14.0 ± 2.0	14.0 ± 2.9	0.0	[−1.17; 1.13]	1.000	0.00	14.0 ± 1.9	−0.0	[−0.76; 0.75]	1.000	0.00	0.0	[−0.99; 1.01]	1.000	0.00
RF—Stiffness (N/m)	273.9 ± 44.1	274.7 ± 69.5	0.8	[−32.65; 34.31]	1.000	0.00	271.1 ± 44.4	−2.8	[−20.28; 14.72]	1.000	0.00	−3.6	[−32.27; 25.05]	1.000	0.00
RF—Relaxation (m/s)	39.2 ± 72.7	22.4 ± 5.0	−16.8	[−60.75; 27.18]	0.975	0.03	22.4 ± 4.4	−16.8	[−60.56; 26.97]	0.968	0.03	−0.1	[−1.49; 1.46]	1.000	0.00

Abbreviations: MAS, Modified Ashworth Scale; T0–T1–T2, time pre–after–30 min after the intervention; SD, standard deviation; CI, confidence interval; ŋ^2^, effect size; *p*, level of significance; (°), degrees; GM, gastrocnemius medialis; GL, gastrocnemius lateralis; RF, rectus femoris.

## Data Availability

Harvard Dataverse—https://doi.org/10.7910/DVN/URWVTC.

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
