# Peer review of "Immediate Effects of TECAR Therapy on Gastrocnemius and Quadriceps Muscles with Spastic Hypertonia in Chronic Stroke Survivors: A Randomized Controlled Trial"

_biomedicines, 2023, doi:10.3390/biomedicines11112973_

Round 1
Reviewer 1 Report
Comments and Suggestions for Authors
The paper presents the assessment of a specific therapy combination consisting of TECAR and functional massage for stroke survivors applied on Gastrocnemius and Quadriceps muscles with Spastic Hypertonia. The paper presents some initial results regarding the approach and should be developed in the future, as the authors have already acknowledged. The methodology is well explained and seems sound. Putting aside the few shortages regarding the small number of patients and the short treatment time, the results seem promising.
The abstract is well written, but the Background should perhaps refer to the extent of the targeted disease to shortly explain the overall state-of-the-art in the field.
The authors mention on page 5 that “Of 44 volunteers (16 females and 28 males) recruited, 2 females and 6 males did not meet the inclusion or exclusion criteria.” So they have been excluded on what terms?
Since the authors have decided to explain the abbreviations of Table 1, perhaps they should also include BMI.
Apparently, the control group has performed slightly more physiotherapy sessions than the experimental group. But it didn’t help too much.
In lines 256-257 the fonts are different.
Some of the references are a bit old, 1990’s and 2000’s, perhaps these should be checked.
Author Response
Reviewer 1
The paper presents the assessment of a specific therapy combination consisting of TECAR and functional massage for stroke survivors applied on Gastrocnemius and Quadriceps muscles with Spastic Hypertonia. The paper presents some initial results regarding the approach and should be developed in the future, as the authors have already acknowledged. The methodology is well explained and seems sound. Putting aside the few shortages regarding the small number of patients and the short treatment time, the results seem promising.
Thank you very much for your comment.
The abstract is well written, but the Background should perhaps refer to the extent of the targeted disease to shortly explain the overall state-of-the-art in the field.
Thank you very much for your comment.
It was added: There is lack of high-quality evidence for many modalities of non-pharmacological inter-ventions for spasticity, with the reference “Khan F, Amatya B, Bensmail D, Yelnik A. Non-pharmacological interventions for spasticity in adults: An overview of systematic reviews. Ann Phys Rehabil Med. 2019 Jul;62(4):265-273. doi: 10.1016/j.rehab.2017.10.001. Epub 2017 Oct 16. PMID: 29042299” and “Hu G, Zhang H, Wang Y, Cong D. Non-pharmacological intervention for rehabilitation of post-stroke spasticity: A protocol for systematic review and network meta-analysis. Medicine (Baltimore). 2021, 7, 100(18):e25788, doi: 10.1097/MD.0000000000025788.
The authors mention on page 5 that “Of 44 volunteers (16 females and 28 males) recruited, 2 females and 6 males did not meet the inclusion or exclusion criteria.” So they have been excluded on what terms?
Thank you for your comment. This information has been added to the text.
“Two participants scored > 2, and one scored < 1 on the Modified Ashworth Scale. One participant carried osteosynthetic material that was incompatible with Tecar Therapy. The remaining three excluded participants were in the subacute phase of stroke.”
Since the authors have decided to explain the abbreviations of Table 1, perhaps they should also include BMI.
Thank you very much for your comment. It was included.
Apparently, the control group has performed slightly more physiotherapy sessions than the experimental group. But it didn’t help too much.
Thank you for your comment. These physiotherapy sessions were the ones that the patients did regularly in their daily lives. But not the study intervention, which was the same for both groups except for the dose of Tecar therapy. Similarly, there was no statistically significant difference between the groups for this descriptive variable.
In lines 256-257 the fonts are different.
Thank you very much for your comment. It was corrected.
Some of the references are a bit old, 1990’s and 2000’s, perhaps these should be checked.
Thank you very much for your comment.
We have changed the reference 24 with Ansari NN, Naghdi S, Arab TK, Jalaie S. The interrater and intrarater reliability of the Modified Ashworth Scale in the assessment of muscle spasticity: limb and muscle group effect. NeuroRehabilitation. 2008;23(3):231-7. PMID: 18560139.
The reference 38 for Cabanas-Valdés, R.; Calvo-Sanz, J.; Serra-Llobet, P.; Alcoba-Kait, J.; González-Rueda, V.; Rodríguez-Rubio, P.R. The Ef-fec-tiveness of Massage Therapy for Improving Sequelae in Post-Stroke Survivors. A Systematic Review and Me-ta-Analysis. Int. J. Environ. Res. Public Health 2021, 18, doi:10.3390/ijerph18094424.
The references 39,40 for García-Bernal MI, González-García P, Madeleine P, Casuso-Holgado MJ, Heredia-Rizo AM. Characterization of the Structural and Mechanical Changes of the Biceps Brachii and Gastrocnemius Muscles in the Subacute and Chronic Stage after Stroke. Int J Environ Res Public Health. 2023 Jan 12;20(2):1405. doi: 10.3390/ijerph20021405. PMID: 36674159; PMCID: PMC9864550.
And the reference 5 was changed for “Peripheral Mechanisms Contributing to Spasticity and Implications for Treatment Antonio Stecco, Carla Stecco & Preeti Raghavan Current Physical Medicine and Rehabilitation Reports volume 2, pag-es121–127 (2014)”.
Reviewer 2 Report
Comments and Suggestions for Authors
Dear Authors,
I found your work very interesting, so I would like to congratulate you on your paper.
Just few minor issues should be revised:
Lines 205-206: the part about the informed consent should be moved into the Materials and Methods section
Line 218: "The" instead of "For"
Lines 319-320: rephrase
Line 322 and 328: would you report some kinds of exercise program that would fit well for patients with stroke?
Lines 336-340: you should move the suggestions for future studies after the limitations
Line 343: typing mistake
Comments on the Quality of English Language
The use of English grammar is fine, jut few minor issues must be checked and fixed
Author Response
Reviewer 2
Dear Authors,
I found your work very interesting, so I would like to congratulate you on your paper.
Just few minor issues should be revised:
Thank you very much for your comment.
Lines 205-206: the part about the informed consent should be moved into the Materials and Methods section.
Thank you very much for your comment. It was done
Line 218: "The" instead of "For"
Thank you very much for your comment. It was done
Lines 319-320: rephrase.
It was done. indicate that spasticity reduction is not always accompanied by functional improvement.
Line 322 and 328: would you report some kinds of exercise program that would fit well for patients with stroke?
Thank you very much for your comment.
We have added Ferry B, Compagnat M, Yonneau J, Bensoussan L, Moucheboeuf G, Muller F, Laborde B, Jossart A, David R, Magne J, Marais L, Daviet JC. Awakening the control of the ankle dorsiflexors in the post-stroke hemiplegic subject to improve walking activity and social participation: the WAKE (Walking Ankle isoKinetic Exercise) randomised, controlled trial. Trials. 2022 Aug 16;23(1):661. doi: 10.1186/s13063-022-06545-w. PMID: 35974379; PMCID: PMC9380386.
Lines 336-340: you should move the suggestions for future studies after the limitations.
Thank you very much for your comment. It was done.
Line 343: typing mistake.
Thank you very much for your comment. It was done.